# Work-Related Stress of Polish School Principals during the COVID-19 Pandemic as a Risk Factor for Burnout

**DOI:** 10.3390/ijerph20010805

**Published:** 2023-01-01

**Authors:** Karina Leksy, Mirosław Wójciak, Grzegorz Gawron, Rafał Muster, Kevin Dadaczynski, Orkan Okan

**Affiliations:** 1Institute of Pedagogy, Department of Social Science, University of Silesia, 40-126 Katowice, Poland; 2Department of Digital Economy Research, University of Economics, 40-287 Katowice, Poland; 3Institute of Sociology, Department of Social Science, University of Silesia, 40-007 Katowice, Poland; 4Department of Health Science, Fulda University of Applied Sciences, 36039 Fulda, Germany; 5Center for Applied Health Science, Leuphana University Lueneburg, 21335 Lueneburg, Germany; 6Department of Sport and Health Sciences, Technical University Munich, 80992 Munich, Germany

**Keywords:** work-related stress, perceived helplessness, perceived self-efficacy, COVID-19 pandemic, Polish school principals, burnout

## Abstract

Background: The ongoing COVID-19 pandemic has influenced educational systems worldwide. School principals coped with numerous significant challenges regarding school management during the epidemiological crisis that could generate a lot of work-related stress. Thus, the presented study examines Polish school principals’ perceived stress and its association with exhaustion and psychosomatic complaints as burnout risk indicators. Principals’ gender and age as sociodemographic control variables were also considered in this paper. Methods: A cross-sectional online study was conducted in eight provinces of Poland from June to December 2021. The study was part of a global COVID-HL school principal survey under the global *COVID-Health Literacy Research Network*. Two subscales of the Perceived Stress Scale (PSS-10) (perceived helplessness [PH] and perceived self-efficacy [PSE]) were considered independent variables in relation to school principals’ mental and physical exhaustion and psychosomatic complaints. Regression models consisting of two equations were used to test the relationship between variables. The first equation consists of the control variables (age, gender), and in the second equation, the independent variables (PH and PSE) were included in addition to the control variables. Results: Almost 50% of school principals experienced a lack of control that caused anger and stress. Mental and physical exhaustion during the pandemic was often or always felt by 30% of respondents. Nearly half of Polish school principals experienced psychosomatic complaints in the form of muscle pain and headaches. PH, to a greater extent than PSE, was associated with mental and physical exhaustion and psychosomatic complaints. With age, the level of psychosomatic complaints and mental and physical exhaustion decreases, but it was higher among women. Regression analysis revealed significant associations between exhaustion and mental health outcomes, even after controlling for demographic variables Conclusion: This study showed that almost half of Polish school principals indicated a high frequency of perceived stress during the pandemic. PH was more substantially associated with mental and physical exhaustion in younger female principals than PSE. Younger female school principals reported more exhaustion and psychosomatic complaints. This finding should be the baseline information for policymakers to improve the wellbeing of Polish school principals and prevent the risk of burnout.

## 1. Introduction

During the last two and a half years, pandemic restrictions almost halted the world. Social interactions were heavily reduced, and lockdowns, quarantines, and physical distancing measures were put in place. This situation might have contributed to many people’s stress and mental health problems [1]. It also included substantial limitations on the functioning of schools. The survey of 122 UNICEF countries in early March 2022 shows that schools have been closed for 20 weeks and partially closed for an additional 21 weeks [2]. According to UNESCO’s Global Monitoring of School Closures, one in ten countries closed their schools entirely for over 40 weeks [3]. School lockdowns were among the primary prevention measures during the pandemic [4]. When the COVID-19 epidemic appeared in Poland, 23,600 schools (including 14,400 primary schools and 9200 secondary schools) were attended by a total of 4.8 million children and youth (2019/2020 school year). In the 2020/2021 school year, there were 4.9 million students in the education system. After the state of the epidemic was announced on 20 March 2020, the education of children and young people in schools in the school years 2019/2020 and 2020/2021 was organised by ad hoc changes to the educational regulations (remote, or hybrid system). The Minister of Education and Science, in cooperation with the Minister of Health, the Chief Sanitary Inspector, and school superintendents, took steps to create legal and organisational conditions that would enable schools to function in the event of a COVID-19 threat [5].

The COVID-19 pandemic has been a traumatic experience for many people worldwide, especially those infected or susceptible to the infection. The adverse impact of the COVID-19 pandemic is relatively unknown [6,7]. Due to school closures, remote learning, and social isolation, children and adolescents experience loneliness, anxiety, and stress [8,9,10,11]. Simultaneously, special attention has been paid to pupils’ psychological well-being during COVID-19, while less consideration has been devoted to other members of the school community [12]. This holds especially true for school principals who are responsible for general school matters and who faced many challenges during the pandemic.

School principals coped with numerous significant challenges regarding school management during the epidemiological crisis [4,13]. They were responsible for implementing policy measures such as organising remote or hybrid learning, adjusting school work to the national and epidemiological requirements, ensuring continuity of education and supporting programmes for students after schools reopened, and ensuring hygiene measures for schools and a safe educational environment [1,14]. The social distancing of teachers and pupils also resulted in additional work and pressure due to the considerable responsibility of school principals. They had to adjust their decisions to the procedures and protocols of the education authorities, which often changed overnight due to the epidemiological situation [15]. School principals had to deal with “managing and coordinating activities of education authorities, teacher, students, and their parents, and to improve student’s performances and teacher’s satisfaction” [15]. All these tasks were much more demanding, considering that there were only limited guides and precedents for managing schools during the pandemic [15]. However, it is worth emphasizing that the Polish Supreme Audit Office positively assesses the activities of the directors of Polish schools, who, despite frequent and urgent changes resulting from the developing epidemic situation, ensured the proper organization and implementation of didactic, educational, and care tasks. Nevertheless, the lack of a systemic approach to the functioning of Polish schools during the epidemic and the related failure to define teaching standards by the Minister was a source of stress for all members of school communities [5].

School principals worldwide work under tremendous pressure since they face many demands starting with filling administrative and management functions such as scheduling, reporting, resolving conflicts, and cooperating with parents and community stakeholders. Another of the school leaders’ tasks is increasing learner achievements, maintaining the satisfaction of teachers, and creating a positive school climate. Motivating teachers and pupils and managing resources effectively to enhance the best educational practices are also essential in the daily work of school principals. Unexpected situations and great crises in the workplace are also their responsibility [16]. The work demands and new conditions of operating faced by school principals during the coronavirus pandemic were, for many, a source of work-related stress [1]. For example, according to the United States nationally representative survey of the well-being of secondary principals (1686 secondary principals) one year after the start of the COVID-19 pandemic, 83% of principals reported experiencing frequent work-related stress during the 2020–2021 school year and this affected women more than men [17]. Fotheringham et al. also surveyed the pressures and influences of the pandemic on school principals during the COVID-19 crisis. Their research revealed that 35% of the participants indicated frequent changes in information were the most problematic. Other challenges, such as a lack of time and clarity of the information received, caused many school principals to perceive difficulties as stressors [18]. Moreover, the awareness of the importance of their role in maintaining the teaching and learning processes at school and the risk of becoming infected could cause fear about their health. The survey conducted by Van Duong et al. proved that symptoms that suggested coronavirus infection raised Taiwan school principals’ concerns about their health and ability to accomplish work [19]. Consequently, those respondents may have experienced more pressure and stress, leading to mental health disorders and burnout. Other results from research conducted among female school principals across Finland revealed that work burnout, workload, COVID-19-related concerns, and difficulty detaching themselves from work are related to the high-stress profile of school principals [1]. Similarly, findings from semi-structured interviews with Norwegian school principals confirmed that increased work demands and the overtime associated with the transformation to remote schooling and the entire digitised educational situation were significant stress factors for both themselves and their staff [20]. Occupational stress during COVID-19 was a result not only for school principals but also for teachers. For example, a study conducted in Ireland revealed moderate or high levels of work burnout for 79% of teachers. The adverse effects of working during the pandemic were reported by teachers in the areas of physical (43%) and mental health (67%), with deterioration in eating (34%), sleeping (70%), and alcohol use (33%) [21]. Simultaneously, many Portuguese teachers reported symptoms of stress, anxiety, and depression [22]. The source of teachers’ burnout was pandemic anxiety and a lack of administrative assistance [23].

Among the different stress conceptualisations, models, and scales [24,25,26,27,28], the transactional concept of stress by Lazarus [29] is the most useful for this study. According to this theory, environmental influences themselves are not stressors. However, they become stressors when an individual perceives them as threats and as something that exceeds the individual’s capacity to deal with [30,31]. Similarly, occupational stress is often a consequence of the mismatch between work resources and what is required (the demands-resources [JD-R] model) [32,33,34]. In this context, resources refer to work conditions within which individual characteristics can be used to attain organisational goals (e.g., control at work and social support; personal resources are self-efficacy, locus of control, skills, and stress coping styles). At the same time, from the standpoint of the job strain model, the most significant risk to physical and mental health disturbances caused by stress occurs to workers facing high psychological workload, demands, or pressures together with low control in meeting those demands [30]. One of the main indicators of the work-related health deterioration process is burnout syndrome [35]. This is defined as “a work-related state of exhaustion that occurs among employees, which is characterised by extreme tiredness, reduced ability to regulate cognitive and emotional processes, and mental distancing. These four core dimensions of burnout are accompanied by depressed mood as well as by non-specific psychological and psychosomatic complaints” [36]. Maslach and Jackson defined the physical and psychological symptoms of burnout and pointed out the loss of self-esteem, depression, alcohol abuse, and exhaustion [37]. Considering that school principals have numerous tasks and occupational responsibilities that might result in high work-related stress, special attention should be paid to this occupational group. As Dadaczynski and Paulus [38] indicated, in comparison to physical health problems, school principals more frequently suffer from mental health problems such as psychosomatic complaints, anxiety, depression, or symptoms of burnout. At the same time, the cited authors pointed out that school principals have been neglected in school health promotion. Meanwhile, it affects all members of a school community, influencing the implementation and overall success of health-promoting activities in the school.

This study examines respondents’ perceived stress and its association with two chosen burnout indicators using quantitative survey methods. We consider perceived helplessness and self-efficacy (two Perceived Stress Scale [PSS-10] subscales) as independent variables potentially associated with school principals’ mental and physical functioning. The COVID-19 pandemic influenced people’s lives, generating a lot of insecurity, anxiety, and fear [39]. Schools worldwide were among the most affected settings [40,41,42], causing school principals to act under unrelenting pressure with limited space to manoeuvre [15]. We assumed this pressure in an already demanding job increased school principals’ likelihood of stress, health problems [43], and risk of burnout. The burnout indicators were: exhaustion and psychosomatic complaints (dependent variables). Emotional exhaustion related to work situations is a significant dimension of burnout that is understood as a consequence of long-term occupational stress [44]. We are aware that the relationship between dependent and independent variables in our regression models could be bidirectional, as many studies have reported the association between exhaustion and work stress [45]. For example, Lau et al. suggested that exhaustion related to work situations could be a predictor of teachers’ perceived stress [46]. In our analysis, principals’ gender and age were introduced as sociodemographic control variables assuming that men and women differ in their perceptions of stress factors [47]. Our findings increase understanding of Polish school principals’ stress levels, mental and physical well-being during the pandemic, and as a result, the potential risk of burnout. Although a large proportion of pandemic research on educational settings was conducted in Poland, its focus is mainly on pupils, teachers, and parents’ opinions concerning remote learning during the coronavirus pandemic [48,49,50]. This is the first community-wide study to examine Polish school principals’ perceived stress levels and their relation to burnout indicators in the middle phase of COVID-19 (June–December 2021). However, it is worth emphasising that some relevant studies have been conducted in other parts of the world concerning school principals’ work during the coronavirus pandemic [19,46,51,52,53,54].

## 2. Materials and Methods

### 2.1. Study Design, Data Collection, and Sample Size

This study on work-related stress among school principals in Poland was conducted as part of an ongoing international study on the global *COVID-Health Literacy Research Network (www.covid-hl.org)*. The survey was conducted online in eight out of 16 provinces in Poland between June 2021 and December 2021. Local government agents from selected provinces (the survey was conducted in the following voivodeships: Śląskie, Podkarpackie, Podlaskie, Kujawsko-Pomorskie, Łódzkie, Warmińsko-Mazurskie, Wielkopolskie, Lubuskie.) responsible for education were informed about the study and asked for help in disseminating the questionnaire among school principals in their regions. The research tool used in this survey and all other country surveys within the COVID-HL Research Network was developed by Dadaczynski et al. [55]. The questionnaire was translated into Polish and sent by email to school principals with an invitation to take part in and complete the online survey. After one month, reminders with anotherinvitation to participate in the research were sent out. Participants were informed about the study’s purpose and importance and, before starting, active consent was required. Completing the questionnaire took about 20–30 min. A total of 1899 Polish school principals took the survey, of which 832 completed the whole questionnaire. Simultaneously, the research sample for individual questions differs because the respondents answered only some of the survey’s questions. Nevertheless, it was a sufficient sample size for analysis.

### 2.2. Measurements

#### 2.2.1. Outcome Variables

Based on the existing burnout tools and interviews with clinical psychologists and psychiatrists [56], a new Burnout Assessment Tool (BAT) [36], also adapted and available in Polish [57], was proposed. It covers the core symptoms (BAT-C) and secondary symptoms (BAT-S) that may occur. BAT-C describes four dimensions: exhaustion, cognitive and emotional impairment, and mental distance. BAT-4S refers to two dimensions: psychosomatic complaints and symptoms of psychological distress constituting secondary symptoms. BAT comprises two distinct scales, of which the first consists of four subscales referring to core symptoms representing burnout, and the second consists of two subscales of secondary symptoms that reflect non-specific symptoms of burnout [57]. The short form of the “Exhaustion” subscale of the BAT inventory is used in this survey [36]. It contains three statements (At work, I feel mentally exhausted; After a day at work, I find it hard to recover my energy; At work, I feel physically exhausted) that were assessed on a 5-point Likert scale from 1 = never to 5 = always. The psychosomatic complaints subscale, which is also part of the BAT inventory [36], contains five items referring to somatic disorders, e.g., headaches, muscle pain, palpitations or chest pain, suffering from stomach or intestinal complaints, and getting sick often. The items were rated on a 5-point Likert scale from 1 = never to 5 = always. Concerning the α value, confirmatory factor analysis was performed to determine content validity. Correlation analyses were used to determine the construct validity. The results indicate that the proposed model structure could not be identified.

#### 2.2.2. Covariates

The PSS-10 [58,59] was used to assess Polish school principals’ work-related stress during the pandemic. It was adapted to the work context during COVID-19. Specifically, the German adaptation by Schneider et al. [59] was verbally adapted to the COVID-19 context (e.g., “…due to the Corona-pandemic”). Additionally, all items have been adapted to the work context (e.g., item 4: “… how often have you felt confident about your ability to handle your professional work-related problems caused by the COVID-19 pandemic?”). The scale consists of ten items rated on a 5-point Likert scale from 1 = never to 5 = very often. One example is “In the last month, how often have you been upset because of something that happened unexpectedly at your work at school?”. The PSS-10 allows the creation of two subscales: (1) Perceived helplessness (PH) (items: 1, 2, 3, 6, 9, 10) and (2) Perceived self-efficacy (PSE) (items: 4, 5, 7, 8). The scores of items 4, 5, 7, and 8 have been reversed. The helplessness concept refers to a belief that nothing can be accomplished to resolve a problem characterized by emotional, motivational, and cognitive deficits. It is a belief that someone’s actions do not influence the action’s outcomes [60]. Individual differences were noticed in the context of developing behaviours characteristic of helplessness. Some findings indicated that up to half of subjects exposed to uncontrollable stress showed no signs of helplessness [61]. Moreover, it was assumed that individual differences in PH are associated with mental and physical health [61].

Self-efficacy is generally defined as people’s belief in their capabilities to achieve different levels of performance attainment [62]. More specifically, it can be understood as one’s belief in their capacity to produce specific performance attainments representing the ability to have positive and negative control over their motivation, behaviour, and social environment [63,64,65]. In this perspective, PSE influences preparation for action because its levels can enhance or impede motivation to behavior change [66]. As well as involving motivation, self-efficacy is also directly related to behaviour enactment, referring to the confidence that one can employ the skills necessary to act, cope with stress, and mobilise the resources required to meet situational demands [67]. Simultaneously, to our knowledge, no studies have reported Polish school principals’ PH and PSE during the pandemic, the impact on their exhaustion, and the psychosomatic disruptions that can lead to burnout.

Additionally, sociodemographic characteristics, including gender and age, were considered the control variables. The age of the respondents has not been operationalised, and the original values were used for analysis.

The KMO (Kaiser-Meyer-Olkin) statistics value for the PSS-10 was 0.863, and for the exhaustion subscale and psychosomatic complaints, respectively: 0.750 and 0.819. These results allowed the use of exploratory factor analysis. All factors based on the Scree-test with eigenvalues >1 were extracted. Two dimensions were obtained for the PSS-10 and one for the exhaustion subscale and psychosomatic complaints. The Cronbach’s alpha coefficient for PH was 0.871 (high scale reliability) and for PSE was 0.649 (acceptable scale reliability). The reliability of the exhaustion subscale was 0.892, and for psychosomatic complaints was 0.777.

### 2.3. Statistical Analysis

Validity verification of the required internal reliability characterising the constructed dimensions was carried out. Exploratory analysis was used to verify the created dimensions and Cronbach’s alpha coefficient to check internal reliability. The study includes the results of frequency, mean values, and standard deviations for thevariables mentioned above. The independence of the control variables, independent variables, and dependent variables were checked. For this purpose, Pearson correlation coefficients ® and the χ^2^ and Mann-Whitney U tests were used.

Regression models consisting of two equations were used to test the relationship between variables. The first equation consists of control variables (age and gender), and in the second equation, independent variables (PH and PSE) were included in addition to the control variables. The F statistic was used to test whether the increase in the coefficient of determination was statistically significant.

The regression models used parameters estimated from standardised parameters, which allow for models without free expression (b0) and for direct comparison of parameter values and the strength of the relationship between different equations. The level of statistical significance was set as a two-sided *p* < 0.05.

All analyses were processed using IBM SPSS Version 28.0 for Windows.

### 2.4. Ethical Consideration

The study was reviewed and approved by the Research Ethics Committee of the University of Silesia in Katowice, Poland (KEUS.118/04.2021). Participants were informed about the aim of the study, its voluntariness, and anonymity.

## 3. Results

### 3.1. Descriptive Statistics

On the item level, the study revealed that 48.5% of the surveyed school principals often and very often felt nervous and stressed at work. Moreover, 47.4% of participants reported being upset due to unexpected events at school and 43.2% of respondents felt angry as things at school were outside of control. Almost 30% of the surveyed principals often felt that difficulties at work were piling up so high that they could not overcome them. The research results also revealed that 26.8% of Polish school principals thought they could not control the important things at school (Table 1).

The presented study also considered the second scale of the PSS-10, PSE (Table 2). In addition to the enormous work pressure during the pandemic, the results obtained proved that most school principals worked at the highest level and felt they did so effectively. For example, 53.8% of school principals were confident about handling professional work-related problems caused by COVID-19. Over half of the respondents (57.2%) stated that things at work at school were ”going their way”. The vast majority of principals (68.8%) were able to control irritations at their work at school.

Data analysis (Table 3) showing the frequency of Polish school principals’ mental and physical exhaustion during the COVID-19 pandemic indicates that over 30% of respondents admitted feeling mentally and physically exhausted at work. Moreover, for nearly half of the surveyed school principals (44.7%), it was hard to recover their energy after a day at work.

Polish school principals most frequently suffered from muscle pain (49.5%) and headaches (27.8%) (Table 4).

Statistical analysis of the PSS-10 subscales showed a higher mean value for a sense of self-efficacy (mean value = 3.56; SD = 0.65) than a sense of helplessness (mean value = 3.15; SD = 0.79). 

According to the descriptive statistics for mental and physical exhaustion, most respondents indicated they “sometimes” experienced exhaustion (mean value = 3.08). Nevertheless, the results were varied (SD = 0.88), and answers ranged from “rarely” (Q25 = 2.33) to “often” (Q75 = 3.67).

The descriptive statistics for psychosomatic complaints show most respondents rarely complain about somatic complaints (mean value = 2.48; SD = 0.78) during the pandemic (Table 5).

### 3.2. Regression Results

In our statistical analysis, two models were performed. The first included the control variables, and the second model contained all the established variables. The first model revealed that the sociodemographic variables are associated with mental and physical exhaustion (F = 8.794 with *p* < 0.001), however, the degree of model explanation is only 2.1% (R^2^ = 0.021). The control variable influencing exhaustion statistically significantly was age (negatively correlated), which means exhaustion decreased with age. Adding the PH and PSE variables to the model described by age and gender makes the level of the related changes in exhaustion increase to 41.1% (R^2^ = 0.411), and this was a statistically significant increase (*p* (ΔR^2^) < 0.001). The parameter itself for the PH variable was positive and statistically significant. Since the standardised parameter value was used in the model, its interpretation follows from the variables’ standard deviations. With this in mind, an increase in the value of the exogenous variable (PH) by the standard deviation value causes an increase in the endogenous variable of exhaustion by SD = 0.638. The parameter standing next to the perceived self-efficacy variable was statistically significant with a negative value (b = −0.100 with *p* = 0.001). Thus, as the value of the PSE variable increases, the value of exhaustion decreases. Comparing the parameter values of the two variables, it can be seen that PSE has a much weaker association with exhaustion than the PH variable (Table 6).

Although there is no gender difference in exhaustion (see Table 5), the *p*-value (=0.052) is very close to the significance, and what is more, statistical analysis revealed differences in both regressions. In the female group, the parameter of PH was 0.581; in the men’s group, it was minimally higher at 0.585. In both groups, its value is similar (difference of 0.004), so it can be said that gender does not play a role in the relationship between PH and exhaustion. In the case of the independent variable of PSE, the parameter’s value is negative in both groups of respondents, being statistically significant (b = −0.095; *p* = 0.004) in the female group, while it tends to be significant in the male group (b = −0.117, *p* = 0.074). Age was associated with exhaustion in both the female and male groups, which means that with a lower age, exhaustion increases. Still, in the female group, it is statistically significant, which means that the level of exhaustion decreases with women’s age (Table 7).

Gender and age were statistically significant in relation to psychosomatic complaints. Negative values of both control variables indicate the level of psychosomatic complaints decreases with age and is higher for women. Control variables describing psychosomatic complaints were statistically significant (F = 30.099 with *p* < 0.001), and the degree of model explanation was 7.1% (R^2^ = 0.071). Adding the PH and PSE variables to the described set of control variables results in an increase in psychosomatic complaints to 28.1% (R^2^ = 0.281), and it is statistically significant (*p* (ΔR^2^) < 0.001). The PH variable was less associated with psychosomatic complaints than exhaustion (the parameter value is lower by almost 0.143—b = 0.638 and b = 0.495) (Table 8).

No association was found between gender and PH and psychosomatic complaints. Age is associated with psychosomatic complaints in both groups, but in the female group, it is statistically significant. In other words, with women’s age, psychosomatic complaints decreased (Table 9).

## 4. Discussion

The pandemic has altered the nature of school principals’ work. They had to extend their roles to create safe educational settings, provide tools and support for virtual teaching, and answer the school community’s concerns and worries [68]. The unexpected and radical changes in working conditions set new expectations and role requirements for principals, many of whom experienced the pandemic as a significant stress [20]. Therefore, this article presents empirical evidence for perceived stress among school principals in Poland (concerning the PH and PSE subscales), its association with mental and physical exhaustion, and psychosomatic complaints (as burnout indicators) during COVID-19. It has been proved that stress negatively affects mental health [37], and helplessness is perceived as an essential component of psychopathological symptoms [61]. In contrast, self-efficacy is a factor that alleviates the effects of stressors on psychological functioning. Moreover, assuming that men and women differ in their perceptions of stress factors [47] and that age is a relevant analytical dimension since it involves diverse expectations and obligations for individuals within the life course [69], in this study, these sociodemographic factors were included as control variables. Additionally, one of the primary goals of the presented research was to assess the burnout risk of Polish school principals potentially caused by working during the demanding and challenging times of COVID-19.

The results obtained proved that almost half of the surveyed school principals often felt nervous and stressed at work (48.5%) and were upset due to unexpected events at school (47.4%). Over 40% of Polish school principals experienced a lack of control that caused anger and stress. Our findings align with research outcomes already mentioned [1,19,20,21,22], confirming the high-stress level of school principals during the pandemic.

School principal self-efficacy has been defined as the principal’s perception of their capacity to fulfil the cognitive and behavioural functions required to arrange the group processes to achieve the school’s goals [70]. Concerning the results presented in this paper, more than half of the surveyed principals manifested a high level of self-efficacy during the COVID-19 pandemic. Overall, school principals in Poland more often indicated self-efficacy than helplessness. We assume they had particular resources (e.g., social support; more decision latitude, and higher salary compared to teachers) that helped them work during the health crisis with conviction about their ability to manage their school efficiently [1]. The statistical analysis also revealed that the increase in Polish school principals’ self-efficacy was related to a decrease in physical and mental exhaustion. We assume that the respondents’ high self-efficacy influenced their adaptation strategies and ability to cope during the pandemic [71]. However, the association between PSE and physical and mental exhaustion was weaker compared to PH. At the same time, there was no association between perceived self-efficacy and psychosomatic complaints. A strong sense of self-efficacy is crucial for school principals as they are more determined to achieve their goals, more flexible, willing to adapt to changing situations [72,73] and manage better in high-demand and high-control conditions [74]. In contrast, principals with a weak sense of self-efficacy are reported to prefer extrinsic or institutional power, experience more anxiety and stress [75], and suffer physical exhaustion and a sense of despair. The level of self-efficacy is one of the more critical issues for school principals, as burnout symptoms are associated with this factor [37].

In the presented study, the regression analysis revealed that Polish school principals’ PH was associated with their physical and mental exhaustion and psychosomatic complaints. Simultaneously, it must be emphasised that helplessness was a stronger predictor for exhaustion than psychosomatic complaints. These findings are consistent with the study of Gmelch and Gates [76], who concluded that there are moderate to high correlations between principals’ emotional exhaustion and stress levels (of which helplessness could be one of the dimensions [47]). Moreover, according to Kirchner et al.’s [77] research results, subjectively PH was a significant predictor for posttraumatic symptom severity (PTSS). The result obtained is also crucial for the risk of burnout, as exhaustion is the core symptom of burnout [57]. Simultaneously, the stressful work conditions of school principals during COVID-19 were potentially detrimental to their health conditions [78]. Many surveys have proven the relationship between work-related stress and physical health [78,79,80], often manifested by psychosomatic symptoms and negatively influencing an individual’s quality of life. Long-term health outcomes can result in many days off and early retirement due to psychosomatic illness [38,81].

Analysis stratified by gender indicates that the associations found seem more important for female principals. The higher level of self-efficacy among women was related to lower mental and physical exhaustion. In addition, woman’s age differentiated the perception of exhaustion and psychosomatic complaints. Their mental and physical exhaustion and psychosomatic complaints decreased with age. Our explanation for the results obtained is that older female principals have a lower likelihood of role conflict between professional and nonprofessional roles. Sociological research shows that women generally experience more difficulties in the reconciliation of work and family than men [69]. With reference to the results obtained, it is worth recalling that in the studies about working and living conditions among local politicians in Sweden and their experiences with combining political work and family life, work-family conflict is highest among the youngest age groups, especially women [69]. Additionally, women still do unpaid work in the home and care for children and relatives, even when they work full-time. This generates a lot of pressure since they cannot possibly reconcile work and family life [47]. Considering the results obtained, we also assume that older female principals were more experienced, which might influence their perception of the unprecedented situation of the pandemic. As a result of their occupational experience, older women may have had a greater sense of self-efficacy [45], through which they felt less mentally and physically exhausted and, thus, less likely to experience psychosomatic complaints. The significance of the professional experience of school principals was confirmed by some research suggesting that the knowledge gained is crucial for a realistic approach to the problems at school [82], overcoming challenges and consequently enhancing their sense of self-efficacy [83]. These findings are also consistent with the Van Duong et al. study [19], showing that older age was associated with a lower likelihood of depressive symptoms. The cited authors similarly explain that older principals may have more experience in managing and solving work difficulties [19]. Simultaneously, special attention must be paid to younger female principals who were more often mentally and physically exhausted and experienced more psychosomatic complaints.

Identifying feelings of helplessness, mental and physical exhaustion, and psychosomatic complaints among Polish school principals during the pandemic (especially younger females) is essential since these can lead to long-term sickness absences [84], burnout, early retirement, or resignation from their positions. This is particularly important in the Polish educational system, given the increasing problem regarding job vacancies for school principals, the resignation of the principal’s functions, earlier retirement, or disability pension [85,86,87]. The pandemic experiences probably exacerbated this trend. According to the US National Association of Secondary School Principals survey, pandemic working conditions accelerated plans to leave the profession for 45% of the surveyed principals in August 2020 [88]. Thus, there is an urgent need to understand principals’ work-related stressors and identify how policymakers and other stakeholders can support principals’ well-being and improve job performance and retention [17].

In the face of the results obtained, we recommend national policymakers and local leaders consider school principals’ well-being and take actions to mitigate work-related stressors, supporting their health and preventing burnout [17]. Work-related stress can be prevented and managed through individual and organisational strategies [89]. These interventions may increase job satisfaction, well-being, autonomy, and perceived stress at the personal level. At the corporate level, these interventions may improve absence rates due to sickness [90]. It is also crucial to focus on the resources which refer to work conditions, such as control at work and social support, and personal resources such as self–efficacy, locus of control, skills, and stress coping styles [30]. To reduce school principals’ stress and workload, sharing some of the principals’ job responsibilities with co-workers or an administrative team would be supportive. Promoting collegiality and collaboration in principals’ work would help create social capital, supporting principals’ well-being [91]. Coaching and mentoring may also provide social support, help principals feel less isolated, and mitigate the overload school principals may experience during a crisis [75,76]. On the individual level, mindfulness sessions could be a recommended solution to support school professionals in managing and reducing work-related stress. The literature shows mindfulness training decreases occupational stress and burnout [92,93,94,95]. The concept of stress management, defined as the human ability to cope with stressful events and situations, may also be effective in raising school principals’ psychological resilience [96]. Revealing a salutogenic leadership style [97] should also be considered as one of the solutions for supporting and enhancing school principals. The salutogenic leadership style is understood as “the ability to promote teachers’ sense of comprehensibility, manageability, and meaningfulness” [38]. Furthermore, social support, which seems to have a substantial stress preventive effect, especially for women [47], should be an essential factor in limiting the negative impact of job demands [1,34,91] and treated as one way of preventing burnout.

The presented research results should be interpreted with some significant limitations. Firstly, it was a cross-sectional study conducted in the middle of the pandemic (June–December 2021). After the first pandemic “shock” (in the spring of 2020), it is highly probable that school principals acquired new skills and adapted better to the situation. This could have caused a decrease in principals’ stress. It is also not feasible to prove a direct connection between school principals’ working in pandemic conditions and their health. One of the reasons is that it is difficult to assess the long-term health effects caused by experiencing work-related stress, which can also be related to individual lifestyles or coping behaviours [19]. Moreover, the cognitive appraisal of stress and the assessment of workload are very subjective and depend on individuals’ features, for example, problem-solving style [98] or type of behavior pattern [99]. Therefore, it is crucial to examine the multiple demands, resources, and well-being indicators associated with principals’ stress profiles in a further study [78]. Nevertheless, the presented research could be the baseline and starting point for creating mental health promotion and burnout prevention strategies for school principals, whose health and well-being are crucial for effectively managing schools, especially during unexpected crises.

## 5. Conclusions

Our study revealed that almost half of Polish school principals indicated high perceived stress during the pandemic. Moreover, PH was associated with mental and physical exhaustion, especially among younger female principals, who reported more fatigue and psychosomatic complaints. The presented study contributes to the limited literature on school principals’ work-related stress in Poland during COVID-19. Therefore, our findings could be a baseline for policymakers concerning Polish school principals’ well-being and burnout risk and prevention. Multidimensional interventions based on the organisation (e.g., changing the organisation’s culture and work practices; supervisor/peer support), organisation-individual (e.g., skills training for school principals; building co-worker social support), and individual (e.g., relaxation; meditation; cognitive-behavioural approaches to improve coping skills) levels [73] could help reduce school principals’ stress and improve their wellbeing and self-efficacy in school management.

## Figures and Tables

**Table 1 ijerph-20-00805-t001:** Frequeny of the PH of Polish school principals during the COVID-19 pandemic.

Indicate How Often You Felt or Thought a Certain Way Due to the COVID-19 Pandemic. In the Last Month …	Total	
	Often/Very Often	N
how often have you been upset because of something that happened unexpectedly at your work at school?	47.4%	928
how often have you felt you were unable to control the important things at your work at school?	26.8%	927
how often have you felt nervous and “stressed” at your work at school?	48.5%	927
how often have you found that you could not cope with all your work tasks at school?	17.3%	925
how often have you been angered because of things that were outside your control at your work at school?	43.2%	925
how often at your work at school have you felt difficulties were piling up so high you could not overcome them?	29.3%	925

Source: Authors’ research.

**Table 2 ijerph-20-00805-t002:** Frequency of the PSE of Polish school principals during the COVID-19 pandemic.

Indicate How Often You Felt or Thought a Certain Way Due to the COVID-19 Pandemic. In the Last Month …	Total	
	Often/Very Often	N
how often have you felt confident about your ability to handle professional work-related problems caused by the COVID-19 pandemic?	53.8%	927
how often have you felt that things at your work at school were “going your way”?	57.2%	926
how often have you been able to control irritations at your work at school?	68.8%	926
how often have you felt you were on top of things at your work at school?	50.7%	924

Source: Authors’ research.

**Table 3 ijerph-20-00805-t003:** Frequency of Polish school principals’ mental and physical exhaustion during the COVID-19 pandemic.

The Following Statements Are Related to Your Work Situation and How You Experience This Situation	Total	
	Often/Always	N
At work, I feel mentally exhausted.	33.8%	834
After a day at work, I find it hard to recover my energy.	44.7%	834
At work, I feel physically exhausted.	31.5%	834

Source: Authors’ research.

**Table 4 ijerph-20-00805-t004:** Frequeny of Polish school principals’ psychosomatic complaints during the COVID-19 pandemic.

How Often Do You Suffer from the Following Complaints?	Total	
	Often/Always	N
I suffer from palpitations or chest pain.	10%	830
I suffer from stomach and/or intestinal complaints.	19%	834
I suffer from headaches.	27.8%	834
I suffer from muscle pain, for example, in the neck, shoulder, or back.	49.5%	834
I often get sick.	7.3%	834

Source: Authors’ research.

**Table 5 ijerph-20-00805-t005:** Sample size concerning gender, age, and descriptive statistics bivariate analyses of all dependent and independent variables.

	Total	PS		PH		PSE		EXH		PC	
	n = 1030	928		928		927		834		834	
	n (%)		*p*		*p*		*p*		*p*		*p*
**Total**		3.31 ± 0.47		3.15 ± 0.79		3.56 ± 0.65		3.08 ± 0.88		2.48 ± 0.78	
**Age**	51.88 ± 6.55	rho = −0.05		rho = −0.06		Rho = 0.03		rho = −0.12		rho = −0.12	
**Gender**			<0.001		<0.001		0.005		0.052		<0.001
Female	837 (81%)	3.34 ± 0.46		3.21 ± 0.78		3.54 ± 0.62		3.11 ± 0.88		2.57 ± 0.77	
Male	193 (19%)	3.20 ± 0.50		2.91 ± 0.79		3.65 ± 0.74		2.96 ± 0.89		2.13 ± 0.72	

rho—Pearson correlation coefficient; *p* (*p*-value)—significance of test; Source: Authors’ research.

**Table 6 ijerph-20-00805-t006:** Findings of the regression analysis for mental and physical exhaustion.

	Equation 1		Equation 2	
Covariates	b	*p*	b	*p*
Gender	−0.066	0.058	0.030	0.262
Age	−0.128	0.000	−0.086	0.001
Independent Variables				
Perceived helplessness			0.586	0.000
Perceived self-efficacy			−0.100	0.001
Equation evaluation				
R^2^	0.021		0.411	
F	8.794		143.586	
*p* (F)	0.000		0.000	
*p* (ΔR^2^)			0.000	
n	774		773	
k	2		4	

R2—coefficient of determination; F—test for the equality of the coefficient of determination; *p* (ΔR^2^)—significance of test for the equality of the coefficient of determination; n—number of observations; k—number of estimated parameters in the model; Source: Authors’ research.

**Table 7 ijerph-20-00805-t007:** Regression results for exhaustion (dependent variable) and perceived helplessness and self-efficacy (independent variables) divided by gender.

	Gender	b	*p*
Age	Women	−0.096	0.001
Perceived helplessness	Women	0.581	0.000
Perceived self-efficacy	Women	−0.095	0.004
Age	Men	−0.054	0.383
Perceived helplessness	Men	0.585	0.000
Perceived self-efficacy	Men	−0.117	0.074

b—regression coefficient; *p* (*p*-value)—significance of test; Source: Authors’ research.

**Table 8 ijerph-20-00805-t008:** Findings of the regression analysis for psychosomatic complaints.

	Equation 1		Equation 2	
Covariates	b	*p*	b	*p*
Gender	−0.226	0.000	−0.155	0.000
Age	−0.128	0.000	−0.097	0.001
Independent Variables				
Perceived helplessness			0.454	0.000
Perceived self-efficacy			−0.033	0.311
Equation evaluation				
R^2^	0.068		0.281	
F	30.099		80.496	
*p* (F)	0.000		0.000	
*p* (ΔR^2^)			0.002	
n	774		773	
k	2		4	

R2—coefficient of determination; F—test for the equality of the coefficient of determination; *p* (ΔR^2^)—significance of test for the equality of the coefficient of determination; n—number of observations; k—number of estimated parameters in the model. Source: Authors’ research.

**Table 9 ijerph-20-00805-t009:** Regression results for psychosomatic complaints (dependent variable) and perceived helplessness and self-efficacy (independent variables) divided by gender.

	Gender	b	*p*
Age	Women	−0.113	0.001
Perceived helplessness	Women	0.453	0.000
Perceived self-efficacy	Women	−0.047	0.211
Age	Men	−0.050	0.471
Perceived helplessness	Men	0.492	0.000
Perceived self-efficacy	Men	0.014	0.851

b—regression coefficient; *p* (*p*-value)—significance of test; Source: Authors’ research.

## Data Availability

Data are available upon request.

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
