# Peer review of "Work-Related Stress of Polish School Principals during the COVID-19 Pandemic as a Risk Factor for Burnout"

_ijerph, 2023, doi:10.3390/ijerph20010805_

Round 1

Reviewer 1 Report

The article is essential and important and teaches about the function of the manager in a crisis and unpredictable era. It provides many details that indicate a struggle in a concrete period.

At the same time - it is very essential that the article concludes with a statement that frames the importance of the article in the two main fields to which it contributes, beyond the tactical level:

One - as an article that teaches about disenfranchised losses, pains, and transparent struggles - in this case of managers - and a reference should be given to the article that deals with group/community that is 'outside' the recognition processes of pain and victimhood:

Lebel, U., " 'Second Class Loss': Political Culture as a Recovery Barrier? – Israeli Families of Terrorist Casualties and their Struggle for National Honors, Recognition and Belonging", Death Studies, 38(1), 2014, 9-19.

The second contribution of the article: the fact that there is an understanding here that it mark a new management is as a new type of leadership - what is now known as "post-heroic" (rather than heroic) leadership. The new leaders and managers also face quite a few of the crises that they themselves are supposed to solve. And there is no longer a representation which link exclusively between a manager/leader = a hero and not a victim of the situation. This is where you should quote:

Lebel, U. and Ben-Shalom, U., "Military Leadership in Heroic and Post-Heroic Conditions", In: Caforio, G. (ed.), 2018, Handbook of the Sociology of the Military, Springer, Rotterdam, pp. 463-475.   

Author Response

Dear Sir or Madam,

We want to thank you for recognizing our contribution to strengthening the evidence base on the association between work stress and the health of school principals during the Covid pandemic. We also thank you for suggesting potential relevant literature. However, after a closer look, we concluded that it did not fit the topic and the target group. Thus, in future studies, we will try to include suggested studies if they are substantively related to our analyses.

Regards,

Karina Leksy

Reviewer 2 Report

Dear authors,

 Your article about “Work-related stress of Polish school principals during the COVID-19 pandemic as a risk factor for mental health disorders” covers an important topic, which gained relevance throughout the pandemic lockdown and still it is worthwhile learning from it after the pandemic.

I think you have a quality and large dataset. I congratulate you on gathering such a large sample size and it seems to cover quite well the country.

While the topic is relevant and the data seem to be good, I think the paper is far from being ready to be published. In its current form it cannot be accepted for several important reasons which I highlight below. I hope this feedback helps you improve your paper and take it closer to being published.

ABSTRACT

The presentation of results in the abstract is a bit confusing. I recommend you rethink the order in which results are presented. It is also important that to avoid all causal references in the description and interpretation of results, as they are stemming from a cross-sectional design.

For instance, in line 29 the word “dependence” or in line 40 you state “lack of control that caused anger and stressed”, which appears again in the discussion.

The conclusion in the abstract regarding effective school management is not “a conclusion from your study” as effectiveness has not been measured in the study.

INTRODUCTION:

The introduction describes the pandemic restrictions and consequences, as well as school principals’ challenges during lockdown. However, there is no input on stress theory, stress models, demands and resources. There are just a few sentences about the relationship between stressors and physical or mental health. There is a strong need for a literature review that covers previous research on stress, health, and previous evidence on their relationship in general and maybe specific in the education sector.

There’s only one paragraph devoted to the presentation of the central topic of the article in P3…and this is rather confusing.

Format. Some sentences need a calm revisión. Eg.p2 L68-69 “the pandemic experience is represented by”??? P2L78 work school or school work? P2 L 99 “proved that symptoms like COVID-19”?(covid is not a symptom, is it?)

MATERIALS AND METHOD

Outcome variables. The authors present in this section one assumption “that exhaustion and psychosomatic complaints” are connected with mental health and can be either risk factor or indicators of mental health disorders”. This idea should be devoted a detailed description of theoretical links between these concepts and empirical evidence on their relationships, as well as attention should be paid to whether one is an antecedent or a consequence and contradictory arguments on both options.

What is BAT inventory? Please detail acronym.

Why present two alphas for exhaustion and none for psychosomatic complaints?

Covariates. Why not use and report the Perceived self-efficacy scale? This would make the study far more interesting.

STATISTICAL ANALYSIS

The presentation of the statistical analysis plan is very confusing and does not follow standard descriptions in psychosocial or public health studies such as this one. Please rewrite in a more concise and clear way.

The theoretical background supporting the models proposed is absent from the article. In particular, an explanation is needed as to why helplessness is considered the independent variable and not the dependent one? Please justify your choice and assignment to IV or DV in both theoretical and empirical previous research. There is also a need to theoretically justify in the introduction why moderation models are hypothesized (what is the theory and previous empirical evidence behind it?)

RESULTS

Table 1. the results presented are based on a sample larger (N=928) than the sample described in the abstract and method  section (N=832).

In p.7 results on self-efficacy are presented. It is important the authors review themselves the article so that it is consistent throughout before submitting it to a journal. Why are the descriptive data presented for self-efficacy, also discussed in the discussion where it acquires more relevance than helplessness, and still the variable is not considered a covariate for helplessness?

P8. Descriptive results of PSS-10 are confusing, a mean is 3.31 but then the mean of self-efficacy is 33.56, and the median 3.50. In the method description it was stated that the scores vary between 1 and 50. Please clarify.

Please clarify the information in the last paragraph in p8 regarding dependence among variables.

The description of the regression results could be summarized, for instance focusing only on the significant effects. It is better to discuss the results in detail than to thoroughly describe what can be read from the tables.

Moderation results  (p.10). There is no justification in the introduction as to why the authors want to test moderation by age. Further, the results show this moderation to be nonsignificant. And then the authors procede with different regression models dividing the sample by gender and school type…The rationale for the study is not clear upfront and the results implicitly present hypotheses.

Discussion. The discussion presents a lot of new information not directly related to the variables in the study. It is fine to do this a bit, but more focused discussion of the results and how they contribute to theory and practice is needed.

Conclusion

The conclusion should stem from what your study has contributed to the topic. At the moment the conclusion is a presentation of practical implications based on other research or applied research areas, not the results from the study. Please revise.

Overall, I think the data and the topic is worthwhile. But a lot more work is needed to build the case in the manuscript, properly explain the contribution of the paper to previous theory and empirical evidence, and manage to take the message across to the readers. Good luck in your efforts!

Author Response

Dear Sir or Madam,

Thank you very much for your insightful review and valuable comments and insights, which allowed us to revise the article substantively and – as we believe – improve its quality. Our changes are highlighted within the manuscript by using track changes. In the following, we describe our changes in more detail.

The manuscript has been extensively revised, starting from adjusting the title of the manuscript – we have changed "mental health disorder" to "burnout". The modified title of the article is: "Work-related stress of Polish school principals during the COVID-19 pandemic as a risk factor for burnout". We think this title fits better with the content presented in the article analysis.

Reviewer's note (Abstract):

The presentation of results in the abstract is a bit confusing. I recommend you rethink the order in which results are presented. It is also important that to avoid all causal references in the description and interpretation of results, as they are stemming from a cross-sectional design.

For instance, in line 29 the word "dependence" or in line 40 you state "lack of control that caused anger and stressed", which appears again in the discussion.

The conclusion in the abstract regarding effective school management is not "a conclusion from your study" as effectiveness has not been measured in the study.

Answer:

The abstract refers to the article in a synthetic way. We have included the most critical findings in the abstract. We have reworked the conclusions and adapted them to our research findings. We removed word "dependence" from the abstract and the whole article and replaced it with word "association".

Reviewer's note (Introduction):

The Introduction describes the pandemic restrictions and consequences, as well as school principals' challenges during lockdown. However, there is no input on stress theory, stress models, demands and resources. There are just a few sentences about the relationship between stressors and physical or mental health. There is a strong need for a literature review that covers previous research on stress, health, and previous evidence on their relationship in general and maybe specific in the education sector. There's only one paragraph devoted to the presentation of the central topic of the article in P3…and this is rather confusing. Format. Some sentences need a calm revision. Eg.p2 L68-69 "the pandemic experience is represented by"??? P2L78 work school or school work? P2 L 99 "proved that symptoms like COVID-19"?(covid is not a symptom, is it?)

Answer:

We have extended and improved the Introduction, including stress theory, models, work-related stress, burnout, and the relationship between stressors and physical and mental health. We also added examples of other research results concerning school principals' work during the coronavirus. We detailed the article's purpose and highlighted the lack of this type of research in Poland. We also justified the control variables - age and gender - used in the study. We revised sentences that were not clear or confusing.

Reviewer's note (Materials and Methods):

Outcome variables. The authors present in this section one assumption "that exhaustion and psychosomatic complaints" are connected with mental health and can be either risk factor or indicators of mental health disorders". This idea should be devoted a detailed description of theoretical links between these concepts and empirical evidence on their relationships, as well as attention should be paid to whether one is an antecedent or a consequence and contradictory arguments on both options. What is BAT inventory? Please detail acronym.Why present two alphas for exhaustion and none for psychosomatic complaints? Covariates. Why not use and report the Perceived self-efficacy scale? This would make the study far more interesting.

Answer:

In the method section, we described all the variables presented in the article. We also justified the choice of independent and dependent variables and pointed out that the relationship between variables can be bidirectional. As the Reviewer suggested, we explained the meaning of the BAT inventory and its acronym. Regarding Cronbach's alphas for exhaustion, we have kindly informed you that in the reviewed article, it already had been provided for perceived helplessness, exhaustion, and psychosomatic complaints. We added Cronbach's alpha for perceived self-efficacy, as this variable was included in the regression analysis.

Reviewer's note (Statistical analysis):

The presentation of the statistical analysis plan is very confusing and does not follow standard descriptions in psychosocial or public health studies such as this one. Please rewrite in a more concise and clear way. The theoretical background supporting the models proposed is absent from the article. In particular, an explanation is needed as to why helplessness is considered the independent variable and not the dependent one? Please justify your choice and assignment to IV or DV in both theoretical and empirical previous research. There is also a need to theoretically justify in the Introduction why moderation models are hypothesized (what is the theory and previous empirical evidence behind it?

Answer:

We have significantly shortened the statistical analysis section, showing the most relevant aspects of the methodology used in the study. This part of the article has been rewritten more concisely. In the methodology section, we explained the differences in the study sample (L. 241-243). We removed the word 'dependence' throughout the article and replaced it with "association". We have revised the regression results and highlighted the most important results. We have removed the moderation analysis and results altogether. We introduced perceived self-efficacy as an independent variable alongside perceived helplessness and checked the regression results for both independent variables.

Reviewer's note (Results)

Table 1. the results presented are based on a sample larger (N=928) than the sample described in the abstract and method section (N=832). In p.7 results on self-efficacy are presented. It is important the authors review themselves the article so that it is consistent throughout before submitting it to a journal. Why are the descriptive data presented for self-efficacy, also discussed in the discussion where it acquires more relevance than helplessness, and still the variable is not considered a covariate for helplessness? P8. Descriptive results of PSS-10 are confusing, a mean is 3.31 but then the mean of self-efficacy is 33.56, and the median 3.50. In the method description it was stated that the scores vary between 1 and 50. Please clarify. Please clarify the information in the last paragraph in p8 regarding dependence among variables. The description of the regression results could be summarized, for instance focusing only on the significant effects. It is better to discuss the results in detail than to thoroughly describe what can be read from the tables. Moderation results (p.10). There is no justification in the Introduction as to why the authors want to test moderation by age. Further, the results show this moderation to be nonsignificant. And then the authors procede with different regression models dividing the sample by gender and school type. The rationale for the study is not clear upfront and the results implicitly present hypotheses.

Answer:

In the method section, we explained the differences in sample size (L. 2014-217). We have written:

The 1899 Polish school principals took the survey, with which 832 completed the whole questionnaire. Simultaneously, the research sample in individual questions differs because the respondents answered only part of the questionnaire questions. Nevertheless, it was a sufficient sample size for the analysis.

We revised the discussion entirely, focusing on the research results. We showed the research results' application dimension and its theoretical and practical contribution. We corrected the descriptive scores of the PSS-10 and the mean value for self-efficacy (mean=3.56). 

Regards,

Karina Leksy

Reviewer 3 Report

Thank you for allowing me to review the manuscript "Work-related stress of Polish school principals during the COVID-19 pandemic as a risk factor for mental health disorders". 

Following my comments:

Abstract

I think the abstract is too long. I would recommend that it be shortened and include a maximum of 2-3 sentences for each section (Background, Methods, Results and Conclusion)

Introduction: 

- The authors present the covid-19 scenario and the difficulties faced by schools. I suggest that the authors also include specific information about the context of Poland to understand what measures schools have put in place to meet the covid-19 requirement.

- The authors should better clarify the concept of helplessness (and other constructs used in the paper) with more literature. 

- In the past two years, much has been written about mental health and covid-19. It is unclear what gap the authors want to fill.

- The goal of the study is unclear. What do the authors mean by "dependence" between perceived helplessness and mental and physical exhaustion and psychosomatic complaints? I believe the authors are referring to the relationship/effect. Perhaps, I would avoid using "dependence." 

- The authors should detail the hypotheses 

Methods: 

- The authors state that the survey took place between June 2021 and December 2021. It is important to clarify the epidemiological situation and measures of the school at that time in Poland.

- Is the BAT inventory validated in Polish?

Results and conclusion

- Some results are presented without explaining. It is important that the authors carefully discuss all the results obtained (also in relation with the hypothesis). For example: (a) With age the level of psychosomatic complaints and exhaustion decreases and is lower for women than men. (b) The type of school does not affect mental and physical exhaustion and psychosomatic complaints. Can the authors try to provide their explanation for this evidence?

- A discussion of intervention practices is needed

Author Response

Dear Sir or Madam,

Thank you very much for your insightful review and valuable comments and insights, which allowed us to revise the article substantively and – as we believe – improve its quality. Our changes are highlighted within the manuscript by using track changes. In the following, we describe our changes in more detail.

The manuscript has been extensively revised, starting from adjusting the title of the manuscript – we have changed "mental health disorder" to "burnout". The modified title of the article is: "Work-related stress of Polish school principals during the COVID-19 pandemic as a risk factor for burnout". We think this title fits better with the content presented in the article analysis.

Reviewer's note (Abstract)

I think the abstract is too long. I would recommend that it be shortened and include a maximum of 2-3 sentences for each section (Background, Methods, Results and Conclusion)

Answer:

As the Reviewer suggested, we have shortened the length of the abstract. The abstract now contains basic information relating to the background, methods, results, and conclusion.

Reviewer's notes (Introduction):

The authors present the covid-19 scenario and the difficulties faced by schools. I suggest that the authors also include specific information about the context of Poland to understand what measures schools have put in place to meet the covid-19 requirement.

Answer:

In the Introduction, we included information on the pandemic situation in Poland and measures taken in schools during the period when the research was carried out. We also added examples of other research results concerning school principals' work during the coronavirus. We detailed the article's purpose and highlighted the lack of this type of research in Poland.

The authors should better clarify the concept of helplessness (and other constructs used in the paper) with more literature.

Answer:

We have extended and improved the Introduction, including stress theory, models, work-related stress, burnout, and the relationship between stressors and physical and mental health. In the method section, we described all the variables presented in the article.

In the past two years, much has been written about mental health and Covid-19. It is unclear what gap the authors want to fill.

Answer:

We detailed the article's purpose and emphasized the lack of research on how school principals coped with a pandemic in Poland. In conclusion, we also indicated that this study contributes to the limited literature on school principals' work-related stress in Poland during the pandemic. Moreover, our findings could be a baseline for policymakers concerning Polish school principals' well-being and burnout risk and prevention.

The goal of the study is unclear. What do the authors mean by "dependence" between perceived helplessness and mental and physical exhaustion and psychosomatic complaints?
I believe the authors are referring to the relationship/effect. Perhaps, I would avoid using "dependence."

Some results are presented without explaining. It is important that the authors carefully discuss all the results obtained (also in relation with the hypothesis). For example: (a) With age the level of psychosomatic complaints and exhaustion decreases and is lower for women than men. (b) The type of school does not affect mental and physical exhaustion and psychosomatic complaints. Can the authors try to provide their explanation for this evidence?

Answer:

We clarified the study goal in the last paragraph of the Introduction. In the discussion section, we carefully showed all relevant research results and gave our insight by suggesting potential explanations. We removed the word 'dependence' throughout the article and replaced it with "association"

Reviewer's notes (Methods):

The authors should detail the hypotheses

Answer:

We deliberately refrained from making hypotheses because we found it challenging to make them about the functioning of school principals in the unprecedented COVID-19 pandemic times. We were guided by cognitive curiosity without focusing on a specific result.

Reviewer's notes (Methods):

The authors state that the survey took place between June 2021 and December 2021. It is important to clarify the epidemiological situation and measures of the school at that time in Poland.

Answer:

Information about the epidemiological situation and measures of the school during COVID-19 in Poland are included in the Introduction.

Is the BAT inventory validated in Polish?

Answer:

We have clarified the meaning of BAT inventory and included 'inventory validated in Polish' in the references.

A discussion of intervention practices is needed

Answer:

We revised the discussion, focusing more on the research results and proposing intervention practices. We have expanded the references in the article to include new bibliographic items.

Regards,

Karina Leksy

Round 2

Reviewer 3 Report

The article appears to be improved from the previous version